# Base-Pairs’ Correlated Oscillation Effects on the Charge Transfer in Double-Helix B-DNA Molecules

**DOI:** 10.3390/ma13225119

**Published:** 2020-11-13

**Authors:** Enrique Maciá

**Affiliations:** Departamento de Física de Materiales, Facultad CC. Físicas, Universidad Complutense de Madrid, E-28040 Madrid, Spain; emaciaba@fis.ucm.es

**Keywords:** DNA charge transfer, effective Hamiltonians, renormalization techniques

## Abstract

By introducing a suitable renormalization process, the charge carrier and phonon dynamics of a double-stranded helical DNA molecule are expressed in terms of an effective Hamiltonian describing a linear chain, where the renormalized transfer integrals explicitly depend on the relative orientations of the Watson–Crick base pairs, and the renormalized on-site energies are related to the electronic parameters of consecutive base pairs along the helix axis, as well as to the low-frequency phonons’ dispersion relation. The existence of synchronized collective oscillations enhancing the π-π orbital overlapping among different base pairs is disclosed from the study of the obtained analytical dynamical equations. The role of these phonon-correlated, long-range oscillation effects on the charge transfer properties of double-stranded DNA homopolymers is discussed in terms of the resulting band structure.

## 1. Introduction

In physiological conditions, the DNA double helix exhibits a full-fledged three-dimensional (3D) geometry, where every two consecutive Watson–Crick base pairs (bps) stand nearly parallel to each other, and they are twisted by a certain angle ($\theta_{0}≃36$${}^{\circ }$ in equilibrium conditions) around the helix axis. In their pioneering work, Eley and Spivey pointed out that a double-stranded DNA (dsDNA) molecule might behave as a one-dimensional (1D) aromatic crystal displaying a π-π based electrical conductivity along the helical axis [1]. The reasoning behind this proposal was that dsDNA’s nucleobases adenine (A), guanine (G), cytosine (C), and thymine (T) are aromatic compounds whose atomic p${}_{z}$ orbitals perpendicular to the plane of the base can form rather delocalized π bonding and $\pi^{*}$ antibonding molecular orbitals. If the orbital overlap between the bps is strong enough, this could lead to extended electronic states along the helical axis, thereby promoting charge transfer (CT) between consecutive bps in an efficient way over long distances through the aromatic bp stack within the DNA helix. Accordingly, CT depends on the intimate coupling among stacked bases, as determined by their relative separation and twist angle, and hence, any perturbation in that stacking, altering the optimal overlapping face-to-face configuration ($\theta_{0}=0$), will significantly affect DNA charge migration. Consequently, one expects structural fluctuations to be an important factor, influencing charge carriers’ transport through dsDNA molecules [2,3,4]. In fact, at physiological temperatures, the relative orientation of neighboring bases becomes a function of time, thereby modifying their mutual overlapping in an oscillatory way. The motion of bases can either occur in a synchronized manner (normal modes’ propagation) or incoherently. The role of thermal fluctuations on the CT efficiency has been discussed in a number of previous works, where the structural fluctuations of the DNA double helix are described by sampling the initial angular velocities and twist angles from a Boltzmann distribution at a given temperature [5,6,7,8,9,10,11,12,13]. Not surprisingly, it was found that the uncorrelated motion of bps, randomly twisting back and forth around the helix axis due to thermal fluctuations, generally reduces π-π stacking overlap, hence degrading the CT efficiency.

On the contrary, the presence of synchronized, collective twist motions of the Watson–Crick bps in DNA duplexes can efficiently enhance the π−π orbital overlapping between non-consecutive bps via a long-range, phonon-correlated tunneling effect [14,15]. In this work, we will focus on coherent charge transport promoted by the coupling between both twist and radial vibration modes and charge motion through duplex DNA, thereby extending previously obtained results [16]. In order to analyze the interplay between the dsDNA low frequency bps’ dynamics and CT efficiency, we will study the coupling between the oscillations of complementary bases along the transversal direction and the twisting motion of each bp as a whole through the helical sugar-phosphate backbone structure, explicitly taking into account its characteristic helical geometry, which has been shown to be very important in biological processes, such as denaturation and transcription [17]. To this end, we will explicitly take into account the stacking interaction, mediated by the orbital overlapping between adjacent bps along the helix, as well as hydrogen bond stretch motions, as described in the Peyrard–Dauxois–Bishop (PDB) model [18,19,20], in the phonon dynamical equations. In our approach, the charge carrier dynamics through a helical dsDNA molecule is expressed in terms of an effective renormalized Hamiltonian describing a diatomic linear chain, where the renormalized transfer integrals explicitly depend on the relative orientations of the Watson–Crick bps and the renormalized on-site energies are related to both the electronic parameters of consecutive codon units along the helix axis, as well as the low-frequency phonon dispersion relation. The corresponding effective hopping terms include both helical and dynamical effects in an intertwined fashion, allowing for a unified treatment of charge-lattice coupled dynamics in a fully analytical way. Thus, we disclose a number of remarkable symmetries of the motion equations themselves, which may be implemented with accurate charge transfer parameters derived from quantum chemistry and molecular dynamics approaches in a straightforward way. Our main conclusion is that a significant improvement of CT can occur in dsDNA via charge-phonon coupling mediated by synchronized helical waves stemming from collective, long-range correlated bps’ oscillation modes.

The paper is organized as follows. In Section 2, we introduce the model Hamiltonian describing the lattice and electronic dynamics. The lattice contribution is expressed in cylindrical coordinates, in order to explicitly take into account the 3D geometry of the double helix DNA molecule. Then, we derive the π−π electronic coupling term describing the molecular orbitals’ overlap through the helix axis in terms of these cylindrical coordinates. This term, describing the transfer integral between successive bps, allows one to relate the dynamical behavior to the CT process by means of a suitable tight-binding electronic model. A convenient feature of our adopted approach is that the resulting 3D fishbone model can be properly mapped into a mathematically simpler effective 1D chain model, still retaining much physico-chemical information in the corresponding renormalization parameters. In Section 3, we obtain the linearized canonical equations of motion for twist and radial lattice variables. In doing so, we introduce a number of characteristic frequencies along with their related time scales. For the sake of simplicity, in Section 3.2, we focus on the dynamics of homopolymer dsDNA molecules, showing the presence of collective oscillations in the form of helical waves. The related dispersion relations for the acoustic and optical branches are analytically derived, and the obtained results are compared to some available experimental results. In this section, we also disclose a very interesting relationship between the dynamics of the dsDNA molecule as a whole and that corresponding to its codon building blocks. In Section 4, we solve the Schrödinger equation for the effective 1D Hamiltonian previously introduced, making explicit use of the helical wave solutions in the transfer integral term. In this way, the charge-phonon coupling effect is fully incorporated in the resulting 3D energy spectrum. Finally, the main conclusions of this work are summarized in Section 5.

## 2. DNA Model Hamiltonian

Two kinds of order coexist in biological DNA, each one related to two separate subsystems in the DNA helix, namely the nucleobase and backbone systems [21]. The informative chemical order determined by the sequence of Watson–Crick bps can be suitably characterized by ab-initio quantum chemistry calculations [22,23], which properly highlight the emergence of molecular orbitals, as is shown in Figure 1b. In order to describe most basic properties of dsDNA molecules, we must consider a model Hamiltonian accounting for different scales of time and space by means of an adequate choice of generalized coordinates including both electronic and dynamic degrees of freedom. According to the Born–Oppenheimer approximation, the lattice and charge dynamics of a dsDNA molecule can be split in terms of the general Hamiltonian $H=H_{l}+H_{e}$, where $H_{l}$ describes the double strand dynamics, and $H_{e}$ describes the CT across nucleobases, as illustrated in the structural lattice model and the electronic tight-binding model depicted in Figure 1a,c, respectively.

### 2.1. Lattice Hamiltonian

In our lattice model, we treat each nucleotide (base + sugar + phosphate) as a point mass, helically arranged and mutually connected by means of elastic rods describing: (1) the sugar-phosphate backbone along a given strand and (2) the interstrand H-bonding between complementary bases (see Figure 1a) [24,25,26]. We explicitly take into consideration the mass difference among the four nucleobases, namely, $m_{G}=347.05$, $m_{C}=307.05$, $m_{A}=331.06$, and $m_{T}=322.05$ amu, and so, we realize that the mass of each bp as a whole is essentially the same, i.e., $M\equiv m_{G}+m_{C}≅m_{A}+m_{T}≅653.5\pm 0.5$ amu. Adopting the reference frame indicated in Figure 1a, the position of the *n*th nucleobase can be expressed as $x_{n}=r_{n}\cos\varphi_{n}$, $y_{n}=r_{n}\sin\varphi_{n}$, and $z_{n}=c\varphi_{n}$, where *n* labels the considered bp along the dsDNA, $r_{n}$ and $\varphi_{n}$ are the usual cylindrical coordinates, and $c=h_{0}/\theta_{0}$, where $h_{0}≃0.34$ nm is the equilibrium distance between two successive bp planes along the *Z* axis in the B-DNA form, while $\theta_{0}$ is the equilibrium relative angular separation between neighboring bps. We note that in this model, the distance between successive bps along the *Z* direction is proportional to the twist angle. In this way, the helical structure is naturally preserved during the dynamical evolution, in agreement with dispersion relation data reported from inelastic X-ray scattering measurements [27]. Thus, we can express the Euclidean distance between adjacent bases along a given strand as:(1)$$d_{n,n\pm 1}=\sqrt{c^{2}\theta_{n,n\pm 1}^{2}+{\left(R_{0}+\rho_{n\pm 1}\right)}^{2}+{\left(R_{0}+\rho_{n}\right)}^{2}-2\left(R_{0}+\rho_{n\pm 1}\right)\left(R_{0}+\rho_{n}\right)\cos\theta_{n,n\pm 1}}$$
where we defined $\theta_{n,n\pm 1}=\pm \left(\varphi_{n\pm 1}-\varphi_{n}\right)$ as the relative angle between two neighboring bps, and $\rho_{n}=r_{n}-R_{0}$ is the radial displacement about the equilibrium position ($R_{0}=1$ nm). We will further assume that the location of the dsDNA as a whole remains fixed, so that the center of mass is constant for each bp. Therefore, the radial displacements about the equilibrium position satisfy the relationship ${\bar{\rho }}_{n}=\lambda_{n}\rho_{n}$, where $\lambda_{n}\equiv m_{n}/{\bar{m}}_{n}$, and henceforth, the upper bar denotes the physical magnitudes of the complementary bases. Therefore, ${\bar{d}}_{n,n\pm 1}$ for the opposite strand can be obtained by simply replacing $\rho_{n}$ with ${\bar{\rho }}_{n}$ in Equation (Equation 1). In equilibrium, both distances reduce to the value $l_{0}={\left.d_{n,n\pm 1}\right|}_{eq.}={\left.{\bar{d}}_{n,n\pm 1}\right|}_{eq.}=\sqrt{h_{0}^{2}+4R_{0}^{2}\sin^{2}\left(\theta_{0}/2\right)}≃0.685$ nm, where we adopted $\theta_{0}=\pi /5.2≃34.6$${}^{\circ }$.

When describing the phonon dynamics of DNA at a molecular scale, one can disregard the inner degrees of freedom of the bases, since we can separate the fast vibrational motions of atoms about their equilibrium positions from the slower motions of molecular groups. Accordingly, we can write the dsDNA molecule lattice Hamiltonian as [16]:(2)$$H_{l}=\frac{1}{2M}\sum_{n=1}^{N}\left(\frac{P_{\rho_{n}}^{2}}{\lambda_{n}}+\frac{P_{\varphi_{n}}^{2}}{\xi^{2}+\lambda_{n}\rho_{n}^{2}+4R_{0}\rho_{n}m_{n}M^{-1}}\right)+U_{H}+U_{S}+U_{B},$$
where *n* runs over the number *N* of bps, $P_{\rho_{n}}$ and $P_{\varphi_{n}}$ are the conjugate momenta of the *n*th bp radial and twist variables, respectively, and $\xi =\sqrt{c^{2}+R_{0}^{2}}≃1.147$ nm is related to the helical geometry of the system, so that $\xi \theta_{n,n\pm 1}$ measures the helix arc length providing the shortest path between two points along a helical coil. In the limit of small radial and twist oscillations ($r_{n}≃R_{0}$, $\theta_{n,n+1}\ll 1)$, Equation (Equation 1) reads $d_{n,n\pm 1}=\sqrt{R_{0}^{2}+c^{2}}\;\theta_{n,n\pm 1}\equiv \xi \theta_{n,n\pm 1}$, so that the Euclidean distance coincides with the helix arc length in this case [14].

The three elastic potential terms in Equation (Equation 2) describe the different interactions between the bases within the framework of the PDB model [18,19], namely:(3)$$U_{H}=\sum_{n=1}^{N}D_{n}{\left[e^{-\frac{\alpha_{n}}{2}\left(1+\lambda_{n}\right)\rho_{n}}-1\right]}^{2},\;$$
represents the radial stretching of the hydrogen bonds connecting complementary bases in the opposite strands of the double helix by means of Morse potentials of depth $D_{n}$ and width $\alpha_{n}$ [19,28]. This potential term includes both the attraction due to the H-bonds forming the bps and the repulsion of the negatively charged phosphates in the backbone of the two strands, which is, in turn, screened by the surrounding solvent water molecules and positively charged counterions. The sequence dependence can be considered by adopting different values for the model parameters $D_{n}$ and $\alpha_{n}$, accounting for the different number of H-bonds in the G≡C and A=T bps [29]. The potential term:(4)$$U_{S}=\frac{1}{8}\sum_{n=1}^{N-1}k_{n,n+1}^{S}\left(1+e^{-\frac{b}{2}u_{n,n+1}^{+}}\right){\left(u_{n,n+1}^{-}\right)}^{2},\;$$
with $u_{n,n+1}^{\pm }=\left(1+\lambda_{n}\right)\rho_{n}\pm \left(1+\lambda_{n+1}\right)\rho_{n+1}$, describes the stacking interaction between adjacent bps, whose role is to inhibit configurations with large relative radial displacements between neighboring pairs. This interaction is characterized by the exponential term that effectively modulates an otherwise harmonic radial oscillation. This term accounts for local constraints in nucleotide motions, measured in terms of the stacking stiffness $k_{n,n+1}^{S}$ and the interaction range *b* parameter, resulting in long-range cooperative elastic effects due to the distortion of hydrogen bonds and the overlap of the π-type orbitals [28]. The description of the radial degree of freedom via the non-linear potentials given by Equations (Equation 3) and (Equation 4) is more realistic than a purely harmonic approach and has been successful in capturing denaturation, as well as transcription initiation processes in several DNA model chains [19,30,31,32]. Finally, the term:(5)$$U_{B}=\frac{k_{B}}{2}\sum_{n=1}^{N-1}\left[{\left(d_{n,n+1}-l_{0}\right)}^{2}+{\left({\bar{d}}_{n,n+1}-l_{0}\right)}^{2}\right],$$
describes the harmonic interaction between neighboring bases along each backbone’s strand.

### 2.2. The π-π Electronic Coupling

The radial and twist oscillations of bps have a significant impact on the molecular orbitals’ overlap throughout the π-stacking, so that the resulting electronic transfer integrals’ values explicitly depend on the dynamical degrees of freedom $\rho_{n}$, $\rho_{n\pm 1}$, and $\theta_{n,n\pm 1}$. The π and $\pi^{*}$ molecular orbitals are formed by the C, N, and O atomic p${}_{z}$ orbitals perpendicular to the bps and pointing along the helical axis, as is illustrated in Figure 2. The p${}_{z}$ orbitals from different bps couple by ppσ>0 and ppπ<0 hybridization, the different signs arising from the respective atomic orbital’s lobe sign. According to the Slater–Koster theory, the transfer matrix element between two p${}_{z}$ orbitals on neighboring bps is given by the combination of ppσ and ppπ hybridization contributions as $V_{ij}$ = $V_{pp\sigma }\sin^{2}\zeta$ + $V_{pp\pi }\cos^{2}\zeta$, where ζ measures the rise angle between successive bps (sinζ = $h_{ij}/d_{ij}$), and: (6)$$V_{ppx}=\eta_{ppx}\frac{\hbar^{2}}{md_{ij}^{2}}\exp\left(-d_{ij}/R_{c}\right)$$
where $\eta_{pp\pi }$ and $\eta_{pp\sigma }$ describe the hybridization matrix elements, *m* is the electron mass, $d_{ij}$= $\sqrt{l_{ij}^{2}+h_{ij}^{2}}$ is the Euclidean distance between atoms belonging to neighboring nucleobases, and $R_{c}$ describes the exponential tails of the atomic wave functions [33,34].

The π-π coupling generally involves one base (say, Y) of the (n+1)th bp and another base (say, X) of the *n*th bp, both belonging to the same strand (i.e., 5’-XY-3’), so that the π-π transfer integral between successive bps along the dsDNA helix is then given by:(7)$$t_{n,n+1}^{XY}=\sum_{i=1}^{N_{1}}\sum_{j=1}^{N_{2}}V_{ij}^{n,n+1}c_{i}^{n+1}c_{j}^{n},$$
where $N_{1}$ and $N_{2}$ are the number of p${}_{z}$ orbitals in the bps n+1 and *n*, respectively, and $c_{j}^{n}$ is the *j*th linear combination of atomic orbitals (LCAO) coefficient of the π molecular frontier orbital (HOMO or LUMO) of bp *n*. Making use of Equation (Equation 6) in Equation (Equation 7), assuming that the mean distances among atoms belonging to different nucleobases can be roughly approximated as $\langle d_{ij}\rangle ≃d$ and $\langle l_{ij}^{2}\rangle ≃l_{n,n\pm 1}^{2}=r_{n}^{2}+r_{n\pm 1}^{2}-2r_{n}r_{n\pm 1}\cos\theta_{n,n\pm 1}$, respectively, we obtain: [14]
(8)$$t_{n,n\pm 1}^{XY}\left(\rho ,\theta \right)=t_{0}^{XY}\left[1-\frac{\bar{\eta }}{d^{2}}\left({\left(R_{0}+\rho_{n}\right)}^{2}+{\left(R_{0}+\rho_{n\pm 1}\right)}^{2}-2\left(R_{0}+\rho_{n\pm 1}\right)\left(R_{0}+\rho_{n}\right)\cos\theta_{n,n\pm 1}\right)\right],$$
where:(9)$$t_{0}^{XY}=\eta_{pp\sigma }\frac{\hbar^{2}}{md^{2}}\exp\left(-d/R_{c}\right)\sum_{i=1}^{N_{1}}\sum_{j=1}^{N_{2}}c_{i}^{n+1}c_{j}^{n},$$
is the transfer integral corresponding to the optimal face-to-face geometry (i.e., $\theta_{n,n\pm 1}\equiv 0$, $\rho_{n}=0$, ∀n) and $\bar{\eta }\equiv 1+\left|\eta_{pp\pi }\right|/\eta_{pp\sigma }=1+2.26/5.27≃1.429$ [35]. In the B-DNA form equilibrium configuration ($\theta_{n,n\pm 1}\equiv \theta_{0}$, $\rho_{n}=0$, ∀n), Equation (Equation 8) reads:(10)$$t_{n,n\pm 1}^{XY}\left(0,\theta_{0}\right)=t_{0}^{XY}\left[1-\bar{\eta }{\left(\frac{2R_{0}}{l_{0}}\sin\frac{\theta_{0}}{2}\right)}^{2}\right].$$

Since $\bar{\eta }>0,$ we get $t_{n,n\pm 1}^{XY}\left(0,\theta_{0}\right)<t_{0}^{XY}$, hence indicating that by explicitly considering helical geometry in the equilibrium configuration, the π−π base coupling strength is significantly reduced below that corresponding to the optimal face-to-face geometry. If we relax the equilibrium structure, allowing for the propagation of low frequency twist oscillations (acoustic modes), though keeping the radial variable describing H-bonding stretch oscillations fixed (no optical modes), Equation (Equation 8) can be approximated as:(11)$$t_{n,n\pm 1}^{XY}\left(0,\theta_{n,n\pm 1}\right)≃t_{0}^{XY}\left(1-\chi \theta_{n,n\pm 1}^{2}\right),$$
for small enough twists ($\cos\theta_{n,n\pm 1}≃1-\theta_{n,n\pm 1}^{2}/2$), where the dimensionless parameter $\chi \equiv \bar{\eta }{\left(R_{0}/l_{0}\right)}^{2}≃2.92$ measures the electron-phonon coupling strength. Despite its approximate nature, Equation (Equation 11) reasonably reproduces the main features of the transfer integral versus twist angle dependence derived from detailed quantum-chemistry calculations in the regime of low energies [12,14]. If we now allow for radial oscillations, still keeping within the small twist angle regime, Equation (Equation 8) adopts the form:(12)$$t_{n,n\pm 1}^{XY}\left(v_{n,n\pm 1},\theta_{n,n\pm 1}\right)≃t_{0}^{XY}\left[1-\chi \left({\left(v_{n}-v_{n\pm 1}\right)}^{2}+\left(1+v_{n}\right)\left(1+v_{n\pm 1}\right)\theta_{n,n\pm 1}^{2}\right)\right],$$
where $v_{n}\equiv \rho_{n}/R_{0}$. Finally, in the limit $v_{n}\ll 1$, Equation (Equation 12) can be approximated as:(13)$$t_{n,n\pm 1}^{XY}\left(v_{n,,n\pm 1},\theta_{n,n\pm 1}\right)≃t_{0}^{XY}\left[1-\chi \left({\left(v_{n}-v_{n\pm 1}\right)}^{2}+\theta_{n,n\pm 1}^{2}\right)\right],$$
where only terms up to the second order are retained.

### 2.3. Electronic Hamiltonian

In order to obtain a realistic description of the rich dsDNA physico-chemistry, keeping at the same time the convenient mathematical simplicity, we will exploit the three-step renormalization approach sketched in Figure 3.

In the fist step, the Watson–Crick bps present in the triplet codon shown in Figure 3a are renormalized to obtain the tight-binding model depicted in Figure 3b. The renormalized on-site energies and transfer integrals are respectively given by [36,37]: $\varepsilon_{ij}=t_{⊥}^{ij}=\varepsilon_{ji}$, which describes the charge carrier hopping from one base to its complementary base within the same bp [38], and: (14)$$\tau_{j}=t_{P}+\frac{\varepsilon_{j}}{t_{P}}\left(E-\gamma_{j}\right),\;\;j=\left\{G,C,A,T\right\},$$
are the effective transfer integrals between the bps and the sugar-phosphate groups, where $t_{P}$ is the glycosidic bond transfer integral, *E* is the charge carrier energy, $\gamma_{j}$ measure the sugar-phosphate groups’ on-site energies, and $\varepsilon_{j}$ are the on-site energies of the corresponding nucleobases. In general, $\gamma_{j}$ will depend on the nature of the neighboring base, as well as the presence of water molecules and/or counterions attached to the backbone. Thus, the renormalized model parameters $\varepsilon_{ij}$ and $\tau_{j}$ entail substantial physicochemical information concerning nucleotide interactions and backbone gating effects [36,39].

We note that the model depicted in Figure 3b can be properly regarded as a 3D generalization of the so-called fishbone model in 2D [40,41]. Accordingly, in the second renormalization step, the sugar-phosphate groups contribution is decimated [36,40,42], so that the original dsDNA molecule is mapped into the equivalent 1D binary lattice shown in Figure 3c, where the renormalized on-site energies labeled α and β correspond to the Watson–Crick complementary bps, and they are explicitly given by ($E\neq \gamma_{j}$):(15)$${\tilde{\varepsilon }}_{n}^{XY}\left(E,\rho ,\theta \right)=t_{n,n\pm 1}^{XY}\left(\rho ,\theta \right)+\frac{\tau_{X}^{2}}{E-\gamma_{X}}+\frac{\tau_{Y}^{2}}{E-\gamma_{Y}}\equiv t_{n,n\pm 1}^{XY}\left(\rho ,\theta \right)+\epsilon_{n}^{XY}\left(E\right),$$
where $t_{n,n\pm 1}^{XY}\left(\rho ,\theta \right)$ accounts for the aromatic base stacking between adjacent nucleotides, given by Equations (Equation 8)–(Equation 13). Therefore, the renormalized on-site potentials ${\tilde{\varepsilon }}_{n}^{XY}\left(E\right)$ explicitly depend on the charge carrier energy, as well as on the angular and radial coordinates describing dsDNA oscillations, thereby enclosing all the relevant physicochemical information of the considered system. In this way, one obtains a realistic description, including 14 electronic model parameters, {$\varepsilon_{j},t_{j},\gamma_{j},t_{⊥}^{GC},t_{⊥}^{AT}$}, fully describing CT throughout dsDNA molecules in terms of just two main functions, namely, $\epsilon_{n}^{XY}\left(E\right)$ and $t_{n,n\pm 1}^{XY}\left(\rho ,\theta \right)$, in a unified way in terms of the effective 1D Hamiltonian:(16)$${\tilde{H}}_{e}^{1D}=\sum_{n=1}^{N}\left(t_{n,n+1}^{XY}\left(\rho ,\theta \right)+\epsilon_{n}^{XY}\left(E\right)\right)c_{n}^{†}c_{n}-\sum_{n=1}^{N-1}t_{n,n+1}^{XY}\left(\rho ,\theta \right)\left(c_{n+1}^{†}c_{n}+c_{n}^{†}c_{n+1}\right),$$
where $c_{n}^{†}$ ($c_{n}$) is the creation (annihilation) operator for a charge at the *n*th site in the chain. In this way, the Hamiltonian given by Equation (Equation 16) provides a realistic treatment of CT mechanisms in dsDNA under physiological conditions, properly taking into account the influence of the dynamical state of the macromolecule on the CT efficiency. It is worth noting that epigenetic processes such as methylation (the addition of a methyl group (-CH${}_{3}$) to one of the bases) will modify the on-site nucleobase energy and its effective mass alike, thereby changing both the mass ratio parameter λ among nucleotides and their γ parameter value. While the role of methylation-related on-site energy changes has been discussed in several recent works [43,44,45], the role of methylation-related dynamical effects in the CT efficiency has not. Due to the presence of both an on-site energy term ($\epsilon_{n}^{XY}\left(E\right)$) and a transfer integral term ($t_{n,n\pm 1}^{XY}\left(\rho ,\theta \right)$) in the diagonal term of the effective Hamiltonian given by Equation (Equation 16), one should expect the possible existence of resonance effects involving both electronic and dynamical physical parameters in an intertwined fashion.

## 3. Dynamical Equations of Motion

### 3.1. General Expressions

From the lattice Hamiltonian given by Equation (Equation 2), we can straightforwardly obtain the canonical equations of motion: $$\left(\xi^{2}+P\left(\rho_{n}\right)\right){\ddot{\varphi }}_{n}=-\frac{k_{B}}{M}\left[\left(1-\frac{l_{0}}{d_{n,n-1}}\right)f_{n-1}-\left(1-\frac{l_{0}}{d_{n,n+1}}\right)f_{n+1}+\left(1-\frac{l_{0}}{{\bar{d}}_{n,n-1}}\right){\bar{f}}_{n-1}-\left(1-\frac{l_{0}}{{\bar{d}}_{n,n+1}}\right){\bar{f}}_{n+1}\right],$$
where $P\left(\rho_{n}\right)=\lambda_{n}\rho_{n}^{2}+\frac{4m_{n}}{M}R_{0}\rho_{n}≃\lambda_{n}\rho_{n}^{2}+2R_{0}\rho_{n}$ and:(17)$$\begin{array}{ccc} f_{n\pm 1} & \equiv & \left(R_{0}^{2}+R_{0}\left(\rho_{n}+\rho_{n\pm 1}\right)+\rho_{n}\rho_{n\pm 1}\right)\sin\theta_{n,n\pm 1}+c^{2}\theta_{n,n\pm 1},\\ {\bar{f}}_{n\pm 1} & \equiv & \left(R_{0}^{2}+R_{0}\left(\lambda_{n}\rho_{n}+\lambda_{n\pm 1}\rho_{n\pm 1}\right)+\lambda_{n}\lambda_{n\pm 1}\rho_{n}\rho_{n\pm 1}\right)\sin\theta_{n,n\pm 1}+c^{2}\theta_{n,n\pm 1}, \end{array}$$
along with: $$\begin{array}{ccc} {\ddot{\rho }}_{n} & = & \frac{D_{n}\alpha_{n}}{M}\frac{1+\lambda_{n}}{\lambda_{n}}\left(e^{-\frac{\alpha_{n}}{2}\left(1+\lambda_{n}\right)\rho_{n}}-1\right)e^{-\frac{\alpha_{n}}{2}\left(1+\lambda_{n}\right)\rho_{n}}\\ & & +\frac{k_{S}}{8M}\frac{1+\lambda_{n}}{\lambda_{n}}\left\{u_{n-1,n}^{-}\left[2+e^{-\frac{b}{2}u_{n,n-1}^{+}}\left(2+\frac{b}{2}u_{n-1,n}^{-}\right)\right]+u_{n+1,n}^{-}\left[2+e^{-\frac{b}{2}u_{n+1,n}^{+}}\left(2+\frac{b}{2}u_{n+1,n}^{-}\right)\right]\right\}\\ & & -\frac{k_{B}}{M}\left[\lambda_{n}^{-1}\left(1-\frac{l_{0}}{d_{n,n-1}}\right)g_{n-1}+\lambda_{n}^{-1}\left(1-\frac{l_{0}}{d_{n,n+1}}\right)g_{n+1}+\left(1-\frac{l_{0}}{{\bar{d}}_{n,n-1}}\right){\bar{g}}_{n-1}+\left(1-\frac{l_{0}}{{\bar{d}}_{n,n+1}}\right){\bar{g}}_{n+1}\right], \end{array}$$
where:(18)$$\begin{array}{ccc} g_{n\pm 1} & \equiv & R_{0}+\rho_{n}-\left(R_{0}+\rho_{n\pm 1}\right)\cos\theta_{n,n\pm 1},\\ {\bar{g}}_{n\pm 1} & \equiv & R_{0}+\lambda_{n}\rho_{n}-\left(R_{0}+\lambda_{n}\rho_{n\pm 1}\right)\cos\theta_{n,n\pm 1}, \end{array}$$
and as a first approximation, we assumed all the stacking stiffness parameters to take on the same value (i.e., $k_{n,n+1}^{S}=k_{S}\;\forall n$). In obtaining Equations (Equation 17) and (Equation 18), we neglected non-linear contributions related to the ${\dot{\varphi }}_{n}{\dot{\rho }}_{n}$ and ${\dot{\varphi }}_{n}^{2}$ terms. Keeping only linear terms in the Taylor series of the functions appearing in the above expressions, we get the linearized equations of motion:(19)$${\ddot{\varphi }}_{n}+\omega_{\varphi }^{2}\left(2\varphi_{n}-\varphi_{n-1}-\varphi_{n+1}\right)=\omega_{\varphi }^{2}l_{B}^{-1}u_{n+1,n-1}^{-},$$
where we reasonably assumed $2R_{0}\rho_{n}\ll \xi^{2}$ and introduced the twist frequency:(20)$$\omega_{\varphi }^{2}\equiv \frac{2k_{B}}{M}{\left(\frac{f_{0}}{\xi l_{0}}\right)}^{2},$$
along with the characteristic length $l_{B}\equiv 2f_{0}/g_{0}$, with $f_{0}=c^{2}\theta_{0}+R_{0}^{2}\sin\theta_{0}≃0.759$ nm${}^{2}$ and $g_{0}=R_{0}\left(1-\cos\theta_{0}\right)≃0.177$ nm, so that $l_{B}≃8.575$ nm (i.e., about 25 bps) and: (21)$${\ddot{\rho }}_{n}+\omega_{\varphi \rho ,n}^{2}\rho_{n}-\frac{1}{2}\omega_{\varphi S,n-1}^{2}\rho_{n-1}-\frac{1}{2}\omega_{\varphi S,n+1}^{2}\rho_{n+1}=\frac{a_{B}}{2}\left(1+\lambda_{n}^{-1}\right)\omega_{\varphi }^{2}\left(\varphi_{n-1}-\varphi_{n+1}\right),$$
where $a_{B}\equiv g_{0}\xi^{2}/f_{0}≃0.307$ nm is a characteristic length whose value is comparable to the equilibrium bps separation $h_{0}$. We introduced the coupled frequencies:(22)$$\omega_{\varphi \rho ,n}^{2}\equiv \frac{{\left(1+\lambda_{n}\right)}^{2}}{\lambda_{n}}\left(\omega_{H,n}^{2}+\omega_{S}^{2}\right)+\frac{1+\lambda_{n}^{2}}{\lambda_{n}}b_{B}^{2}\omega_{\varphi }^{2},$$
and:(23)$$\omega_{\varphi S,n\pm 1}^{2}\equiv \frac{\left(1+\lambda_{n}\right)\left(1+\lambda_{n\pm 1}\right)}{\lambda_{n}}\omega_{S}^{2}-\frac{1+\lambda_{n}\lambda_{n\pm 1}}{\lambda_{n}}b_{B}^{2}\omega_{\varphi }^{2},$$
where $b_{B}\equiv a_{B}/\xi =g_{0}\xi /f_{0}≃0.267$ is a dimensionless factor. Therefore, $\omega_{\varphi \rho ,n}$ depends on the twist frequency, as well as the radial stretch H-bonding and lateral stacking oscillations of bps, whose characteristic frequencies are $\omega_{H,n}^{2}=D_{n}\alpha_{n}^{2}/\left(2M\right)$, and $\omega_{S}^{2}=k_{S}/M$, respectively. The site label in $\omega_{\varphi \rho ,n}$ arises from the presence of the $\lambda_{n}$ factor, as well as the fact that $\omega_{H,n}^{2}$ is site dependent, due to the different Morse potential parameters values for G≡C and A=T bps. On the other hand, $\omega_{\varphi S,n\pm 1}$ involves lateral stacking and twist oscillations. In this case, the site label dependence involves all the $\lambda_{k}$ terms. The set of coupled Equations (Equation 19) and (Equation 21) describes the dynamics of general dsDNA molecules, where two kinds of bps can be arranged either periodically or aperiodically [21,46,47,48,49].

### 3.2. Dynamics of Homopolymer dsDNA Macromolecules

For the sake of simplicity, we will consider in this section the homopolymer case (i.e., polyA-polyT or polyG-polyC chains). In this case, the renormalized chain shown in Figure 3c becomes an effective monoatomic lattice (i.e., α≡β), where the renormalized on-site potentials depend on both the electron energy *E* and the phonon wavevector *q*, due to the presence of the π-π transfer integral in Equation (Equation 15). In addition, $\lambda_{n}=\lambda_{XY}$, $D_{n}=D_{XY}\equiv D$ and $\alpha_{n}=\alpha_{XY}\equiv \alpha$ (see Table 1), so that the frequencies $\omega_{H,n}\equiv \omega_{H}$, $\omega_{\varphi \rho ,n}\equiv \omega_{\varphi \rho }$ and $\omega_{\varphi S,n\pm 1}\equiv \omega_{\varphi S}$ are no longer site dependent. Hence, Equations (Equation 19)–(Equation 21) can be rewritten as (henceforth, we will drop the subscript XY in the λ parameters for the sake of clarity):(24)$${\ddot{\varphi }}_{n}+\omega_{\varphi }^{2}\left(2\varphi_{n}-\varphi_{n-1}-\varphi_{n+1}\right)=A_{\lambda }\omega_{\varphi }^{2}\left(\rho_{n+1}-\rho_{n-1}\right),$$
(25)$${\ddot{\rho }}_{n}+\frac{1}{2}\omega_{\varphi \rho }^{2}\left(2\rho_{n}+\rho_{n-1}+\rho_{n+1}\right)-\frac{1}{2}\omega_{HS}^{2}\left(\rho_{n-1}+\rho_{n+1}\right)=B_{\lambda }\omega_{\varphi }^{2}\left(\varphi_{n-1}-\varphi_{n+1}\right),$$
where $A_{\lambda }\equiv \left(1+\lambda \right)l_{B}^{-1}$ and $B_{\lambda }\equiv a_{B}\left(1+\lambda^{-1}\right)/2$ are constants, and we introduced the frequency $\omega_{HS}^{2}\equiv \mu \left(\omega_{H}^{2}+2\omega_{S}^{2}\right)$, describing the coupling between the stacking and H-bond radial stretch oscillations, where $\mu \equiv \lambda^{-1}{\left(1+\lambda \right)}^{2}$ can take on two values. Making use of the model parameters listed in Table 1, we obtain the values listed in Table 2 for the characteristic frequencies just introduced, along with their related time scales.

From the data listed in Table 2, we see that the time scale of angular motions, determined by the twist frequency, amounts to ∼6 ps, which are an order of magnitude slower than those corresponding to the twist-radial coupled oscillations. The time scale related to bps’ H-bond and stacking motions occupy an intermediate position, whereas coupled oscillations involving stretch and stacking motions are about 2.6 times quicker than the H-bond-mediated stretch oscillations alone. For the sake of comparison, the transition times reported for intrastrand hole transfer in ds-GT${}_{n}$GGG oligonucleotides range from τ=0.5 ps for n=1 to τ=315 ps for n=4 [63]. Quite interestingly, the electrical response of biological dsDNA chains to light irradiation has been recently investigated in order to engineer a DNA based molecular switch. In these experiments, it was observed that the electrical current turns on when the frequency of the incident light is above the 2 THz threshold [64], a value that coincides with that listed for $\nu_{\varphi \rho }$ in Table 2. On the other hand, it is worth mentioning that the $\nu_{HS}$ frequency value listed in Table 2 is smaller than the 2.83 and 3.04 THz frequencies experimentally observed by optical Kerr-effect spectroscopy in ds-GGCGGCCCGCGCGGGCCGCC and ds-ATTATTATTATATTA oligonucleotides, respectively. These oscillations are consistent with delocalized optical phonon modes with a wavelength extending throughout the molecule as a whole, and they are assumed to be related to the H-bonding dynamics between the two strands [65,66,67].

The mathematical structure of Equations (Equation 24) and (Equation 25) clearly indicates the correlated nature of next-neighboring bps dynamics. It is then convenient to zoom out our perspective and focus our attention on the dynamics of consecutive triplets of bps along the double-helix, which are closely related to the so-called codon units in genomics. To this end, we properly add up the dynamical equations corresponding to the consecutive bps corresponding to sites n−1, *n*, and n+1, grouping the resulting expression in terms of the collective variables $x_{n}\equiv 2\varphi_{n}-\varphi_{n-1}-\varphi_{n+1}$ and $y_{n}\equiv 2\rho_{n}-\rho_{n-1}-\rho_{n+1}$, to obtain:(26)$${\ddot{x}}_{n}+\omega_{\varphi }^{2}\left(2x_{n}-x_{n-1}-x_{n+1}\right)=A_{\lambda }\omega_{\varphi }^{2}\left(y_{n+1}-y_{n-1}\right),$$
(27)$${\ddot{y}}_{n}+\frac{1}{2}\omega_{\varphi \rho }^{2}\left(2y_{n}+y_{n-1}+y_{n+1}\right)-\frac{1}{2}\omega_{HS}^{2}\left(y_{n-1}+y_{n+1}\right)=B_{\lambda }\omega_{\varphi }^{2}\left(x_{n-1}-x_{n+1}\right).$$

Remarkably enough, we realize that Equations (Equation 26) and (Equation 27), describing the codon dynamics as a whole, are formally identical to Equations (Equation 24) and (Equation 25), which describe the motion of their constituent bps, since they are invariant upon the simultaneous variable exchange $\varphi_{n}↔x_{n}$ and $\rho_{n}↔y_{n}$. This property can be regarded as expressing a renormalization of the dynamical equations when going from the bp local scale to the longer triplet codon scale. Quite interestingly, the very mathematical structure of the codon dynamics prescribed by Equations (Equation 26) and (Equation 27) guarantees that we will obtain similar dynamical equations by grouping codons in successive triplets of nested codon units recurrently, all the way up to the entire DNA molecule itself. Accordingly, the set of Equations (Equation 24) and (Equation 25) exhibits a self-similar symmetry upon triplet renormalization operation, so that by solving this fundamental dynamical equation set, we are actually disclosing the main features of the dynamics of the entire dsDNA macromolecule as a whole.

Inspired by previous results [14,15], we look for solutions to Equations (Equation 24) and (Equation 25) of the form $\varphi_{n}=\varphi_{0}e^{i(\omega t-nq\xi )}$ and $\rho_{n}=\rho_{0}e^{i(\omega t-nq\xi )}$, describing a helical wave propagating throughout the dsDNA with frequency ω and wave vector *q*, where $\varphi_{0}≃8$${}^{\circ }$=0.14 rad and $\rho_{0}≃0.05$ nm are the twist and radial oscillation amplitudes at ambient temperature, respectively [68]. In so doing, Equations (Equation 24) and (Equation 25) can be expressed in the matrix form:(28)$$\left(\begin{array}{cc} 2\omega_{\varphi }^{2}\left[1-\cos\left(q\xi \right)\right]-\omega^{2} & 2iA_{\lambda }\omega_{\varphi }^{2}\sin\left(q\xi \right)\\ -2iB_{\lambda }\omega_{\varphi }^{2}\sin\left(q\xi \right) & \omega_{\varphi \rho }^{2}-\omega_{\varphi S}^{2}\cos\left(q\xi \right)-\omega^{2} \end{array}\right)\left(\begin{array}{c} \varphi_{0}\\ \rho_{0} \end{array}\right)=\left(\begin{array}{c} 0\\ 0 \end{array}\right).$$

The solution to Equation (Equation 28) requires the matrix determinant to identically vanish, thereby leading to a biquadratic equation whose solutions yield the dispersion relations for the acoustic and optical phonon branches given by:(29)$$\omega_{\pm }^{2}=\frac{1}{2}\left(G(q)+H(q)\right)\pm \frac{1}{2}\sqrt{{\left(H(q)-G(q)\right)}^{2}+16C_{\lambda }\omega_{\varphi }^{4}\sin^{2}\left(q\xi \right)},$$
where $G\left(q\right)\equiv 4\omega_{\varphi }^{2}\sin^{2}\left(q\xi /2\right)$, $H\left(q\right)\equiv \omega_{\varphi \rho }^{2}-\omega_{\varphi S}^{2}\cos\left(q\xi \right)$, and $C_{\lambda }\equiv A_{\lambda }B_{\lambda }$. The exact dispersion relations given by Equation (Equation 29) are plotted in Figure 4, though they can be very well approximated by the simpler expressions:(30)$$\nu_{-}^{2}=4\nu_{\varphi }^{2}\sin^{2}\left(\frac{q\xi }{2}\right)\;\nu_{+}^{2}=\mu \left[\nu_{H}^{2}+2\nu_{S}^{2}\sin^{2}\left(\frac{q\xi }{2}\right)\right],$$
which extend previously reported results [25,30]. As we see, the acoustic branch is completely determined by twist oscillations, whereas the optical branch depends on both stretch and stacking oscillations. In particular, the q=0 bandgap, $\Delta \nu \left(0\right)\equiv \nu_{+}\left(0\right)-\nu_{-}\left(0\right)=\sqrt{\mu }\nu_{H}≃1.70$ THz (≃7 meV), is fixed by the H-bond frequency value. The maximum bandgap width occurs for $q^{*}=\pi /\xi ≃2.732$ nm${}^{-1}$, with $\Delta \nu \left(q^{*}\right)=\nu_{HS}-2\nu_{\varphi }≃1.91$ THz. The sound velocity obtained from the acoustic dispersion curve is 1.2 km s${}^{-1}$, a figure smaller than the experimentally reported values ranging from 1.7 to 4.3 km s${}^{-1}$, depending on the employed technique [69].

An interesting subject that can be addressed within the framework presented in this work refers to the behavior of the correlated oscillations when the DNA molecule is bonded with other small molecules. Generally speaking, the presence of ligands attached onto the sugar-phosphate backbone simultaneously affects the electronic properties and the mass distribution in a local region of the DNA molecule. This twofold effect can be properly accounted for in terms of a change in the γ value parameter, which modifies the electronic bandgap according to Equation (Equation 40) and a change in the values of parameters λ and *M*, thereby modifying the frequency values given by Equations (Equation 20)–(Equation 23). Thus, any effective bp mass increase leads to a slow down of frequencies $\omega_{\varphi }$, $\omega_{H}$, and $\omega_{S}$, along with their related coupled frequencies. In addition, the reduction of the twist frequency value makes the acoustic dispersion relation slope decline, so that the sound speed is reduced as well, ultimately leading to a lower thermal conductivity around the place where the small molecule has been bonded. On the other hand, a smaller $\omega_{\varphi }$ value will widen the gap between the acoustic and optical branches (see Figure 4).

## 4. Charge Transfer through dsDNA Homopolymers

Making use of the helical wave complex expressions for the variables $\varphi_{n}\left(t\right)$ and $\rho_{n}\left(t\right)$ into Equation (Equation 13), we get:(31)$$t_{n,n\pm 1}^{XY}\left(\rho_{n},\theta_{n,n\pm 1}\right)≃t_{0}^{XY}\left[1-4\chi A_{0}^{2}\sin^{2}\left(\frac{q\xi }{2}\right)\right],$$
where $A_{0}^{2}\equiv \varphi_{0}^{2}+{\left(\rho_{0}/R_{0}\right)}^{2}≃0.02$, so that $4\chi A_{0}^{2}≃0.26$ eV and $t_{n,n\pm 1}^{XY}>0,\forall q$. The resulting transfer integrals become site independent, which is a natural consequence of the synchronized motion mediated by helical waves propagating in the dsDNA molecule. Accordingly, we can write $t_{n,n\pm 1}^{XY}\left(\rho_{n},\theta_{n,n\pm 1}\right)=t^{XY}\left(q\right)\equiv t^{XX}\left(q\right)$ in the homopolymer case, and the Schrödinger nearest-neighbor tight-binding equation corresponding to the electronic Hamiltonian given by Equation (Equation 16) reads:(32)$$\left[E-\epsilon^{XY}\left(E\right)-t^{XX}\left(q\right)\right]\psi_{n}-t^{XX}\left(q\right)\left(\psi_{n+1}+\psi_{n-1}\right)=0,$$
where $\psi_{n}$ is the electronic wave function at site *n*. Equation (Equation 32) can be expressed in the matrix form:(33)$$\left(\begin{array}{c} \psi_{n+1}\\ \psi_{n} \end{array}\right)=\left(\begin{array}{cc} \frac{E-\epsilon^{XY}\left(E\right)}{t^{XX}\left(q\right)}-1 & -1\\ 1 & 0 \end{array}\right)\left(\begin{array}{c} \psi_{n}\\ \psi_{n-1} \end{array}\right)\equiv M\left(E,q\right)\left(\begin{array}{c} \psi_{n}\\ \psi_{n-1} \end{array}\right),$$
and within the framework of the transfer matrix formalism, the charge carrier dispersion relation of a dsDNA molecule containing *N* bps is given by (assuming periodic boundary conditions):(34)$$\cos\left(\kappa Nh_{0}\right)=\frac{1}{2}\mathrm{tr}\left[M_{N}\left(E,q\right)\right]\equiv \frac{1}{2}\mathrm{tr}\left[\prod_{n=N}^{1}M\left(E,q\right)\right]=\frac{1}{2}\mathrm{tr}\left[M^{N}\left(E,q\right)\right],$$
where κ is the charge carrier wavevector. Since detM(E,q)=1, we can use the Cayley–Hamilton theorem for unimodular matrices in order to calculate the required power matrix as $M^{N}\left(E,q\right)=U_{N-1}M\left(E,q\right)-U_{N-2}I$, where *I* is the identity matrix and $U_{m}\left(x\right)\equiv \sin\left[(m+1)\phi \right]/\sin\phi$, with x≡cosϕ≡trM/2, are Chebyshev polynomials of the second kind, satisfying the recurrence relationship $U_{m+1}-2xU_{m}+U_{m-1}=0$. In this way, we obtain [21]:(35)$$M^{N}\left(E,q\right)=U_{N-1}\left(\begin{array}{cc} 2x & -1\\ 1 & 0 \end{array}\right)-U_{N-2}\left(\begin{array}{cc} 1 & 0\\ 0 & 1 \end{array}\right)=\left(\begin{array}{cc} U_{N} & -U_{N-1}\\ U_{N-1} & -U_{N-2} \end{array}\right).$$

Taking into account the relationship $U_{m}-U_{m-2}=2\cos\left(m\phi \right)$, the charge carrier dispersion relation can then be expressed as:(36)$$E=\epsilon^{XY}\left(E\right)+t^{XX}\left(q\right)\left(1+2\cos\left(\kappa h_{0}\right)\right).$$

According to Equation (Equation 15), the $\epsilon^{XY}\left(E\right)$ function entails detailed information regarding the electronic structure model of the dsDNA molecule. X-ray experiments indicated that counterions condense around the nucleic acid chain in a tightly-bound layer, in agreement with early model calculations [70]. Therefore, a homogeneous charge distribution through the backbone can be assumed as a first approximation, that is $\gamma_{j}\equiv \gamma$. In that case, we can write:(37)$$\epsilon^{XY}\left(E\right)=2\left[t_{P}^{2}+a^{XY}\left(E-\gamma \right)+b^{XY}{\left(E-\gamma \right)}^{2}\right]{\left(E-\gamma \right)}^{-1},$$
where $a^{XY}\equiv \varepsilon_{X}+\varepsilon_{Y}$ and $b^{XY}\equiv \left(\varepsilon_{X}^{2}+\varepsilon_{Y}^{2}\right)/\left(2t_{P}^{2}\right)$. Thus, plugging Equations (Equation 31) and (Equation 37) into Equation (Equation 36), we obtain the charge carrier dispersion relation in the explicit polynomial form:(38)$$B_{2}^{XY}E^{2}+\left[B_{1}^{XY}+F\left(\kappa ,q\right)\right]E+2B_{0}^{XY}-\gamma F\left(\kappa ,q\right)=0,$$
where $B_{2}^{XY}\equiv 2b^{XY}-1$, $B_{1}^{XY}\equiv \gamma \left(1-4b^{XY}\right)+2a^{XY}$, and $B_{0}^{XY}\equiv \gamma \left(\gamma b^{XY}-a^{XY}\right)+t_{P}^{2}$, and we introduced the auxiliary function:(39)$$F\left(\kappa ,q\right)\equiv t_{0}^{XX}\left[1-4\chi A_{0}^{2}\sin^{2}\left(\frac{q\xi }{2}\right)\right]\left[1+2\cos\left(\kappa h_{0}\right)\right].$$

The resulting energy spectrum structure consists of two slightly asymmetric bands, $E_{\pm }\left(\kappa ,q\right)$, with relatively small widths ($W_{\pm }$), separated by a gap, $E_{g}\left(0,q\right)$, whose value depends on the phonon wavevector *q*, as is illustrated in Figure 5 and Table 3 for the particular case γ=0. By inspecting Figure 5, we can clearly appreciate that the phonon coupling gives rise to a systematic reduction of the bandgap width $E_{g}\left(0,0\right)$ as *q* increases up to the value $q^{*}=\pi /\xi$ (see Figure 4), thereby enhancing the CT efficiency. Thus, the bandgap relative variation amounts to about 9% (6.5%) for polyG-polyC (polyA-polyT), respectively, as compared to the $E_{g}\left(0,0\right)$ value. We note that the bandgap values listed in Table 3 are remarkably smaller than those usually reported in other studies [71,72,73,74,75,76]. Indeed, the dsDNA electronic structure is very sensitive to the precise value of the sugar-phosphate on-site energy. Thus, for the general case γ≠0, the bandgap width can be explicitly expressed as:(40)$$E_{g}\left(0,0\right)=\frac{1}{2b^{XY}-1}\sqrt{{\left(3t_{0}^{XX}-\gamma +2a^{XY}\right)}^{2}-8\left(2b^{XY}-1\right)t_{P}^{2}},$$
so that a semiconductor-semimetal transition can be promoted by properly tuning the adopted γ value [36].

## 5. Conclusions

The physical picture inspiring this work relies on the fact that the presence of collective, orchestrated oscillation motions of bps within the DNA double helix structure can efficiently enhance the π−π orbital overlapping between bps along the backbone chain, hence promoting charge transfer via a long-range, phonon-correlated tunneling effect involving bps, which are relatively far apart. This property is intimately related to the helical geometry of the nucleobases’ arrangement along the duplex chain, which makes possible that a local hopping process, involving a relatively small number of neighboring nucleotides, ultimately extends over the entire DNA chain as a consequence of the synchronized nature of the resulting helical wave. The possible presence of these helical waves may be relevant in the study of CT properties in dsDNA polymers exhibiting extensive chemically homogeneous regions along their strands, such as those reported in genomic studies of the telomere sequences of certain invertebrates [77] or in tandem repeats [78], involving mononucleotide triplet motifs (AAA, TTT, GGG, or CCC) [79]. Indeed, the existence of DNA-mediated charge migration has been related to the understanding of the damage recognition process, including the presence of lesions and mismatches. Furthermore, since CT dependence can be sensed electrically, it can be exploited for nanotechnological applications through base modifications or DNA-protein binding or with the task of designing nanoscale sensing of genomic mutations, opening new challenges for emerging nanobiotechnologies [80,81,82,83]. In addition, the fundamental dynamical mechanisms reported in this work are expected to also take place in other π−π molecular wires, such as G based quadruplexes [84].

## Figures and Tables

**Figure 1 materials-13-05119-f001:**
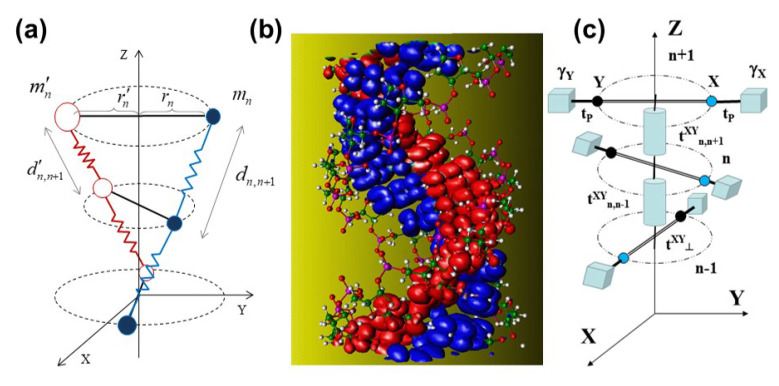
(**a**) Diagram of the dsDNA lattice model showing the harmonic bonds between adjacent bps through the sugar-phosphate backbone (zig-zag lines) and between complementary bases due to H-bonds (solid lines). (**b**) Full-atom dsDNA model with surfaces of constant charge density for the states corresponding to the π-like lowest unoccupied molecular orbital (LUMO, in red) of the C bases and the highest occupied molecular orbital (HOMO, in blue) of the G bases of a polyG-polyC molecule in the dry conditions A form (Courtesy of Emilio Artacho) [23]. (**c**) Diagram of the dsDNA electronic model showing the π-π channel transfer integrals $t_{n,n\pm 1}^{XY}$ (cylindrical rods), the interstrand transfer integrals between complementary bases $t_{⊥}^{XY}$ (perpendicular bars), the glucosidic transfer integrals $t_{P}$ between nucleobases (spheres) and sugar-phosphate groups (cubes), and the on-site energies of these groups $\gamma_{j}$.

**Figure 2 materials-13-05119-f002:**
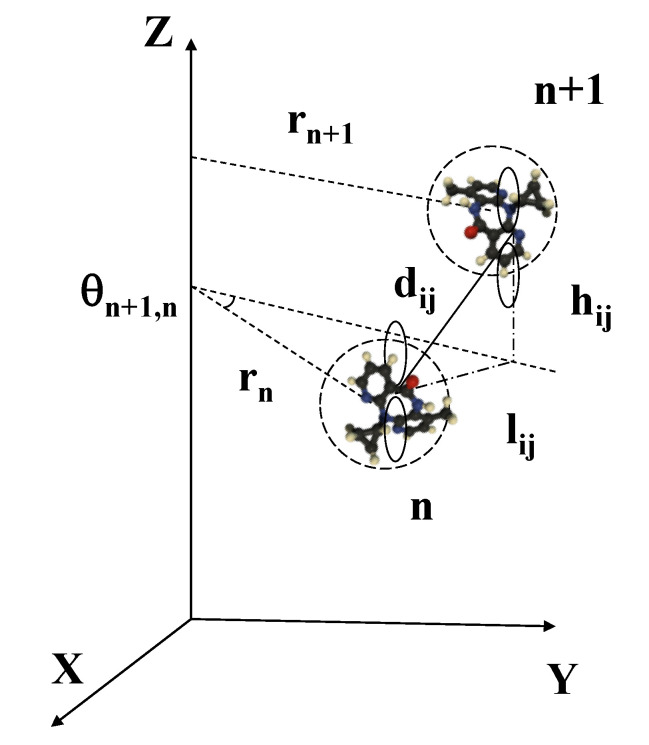
The p${}_{z}$ atomic orbitals overlapping between neighboring nucleobases are described in terms of their Euclidean distance $d_{ij}$, which is determined by the relative twist and radial variables $\theta_{n,n+1}$, $r_{n+1}$, and $r_{n}$. Reprinted figure with permission from Maciá, E. Physical Review B, 76, 245123, 2007. Copyright (2007) by the American Physical Society.

**Figure 3 materials-13-05119-f003:**
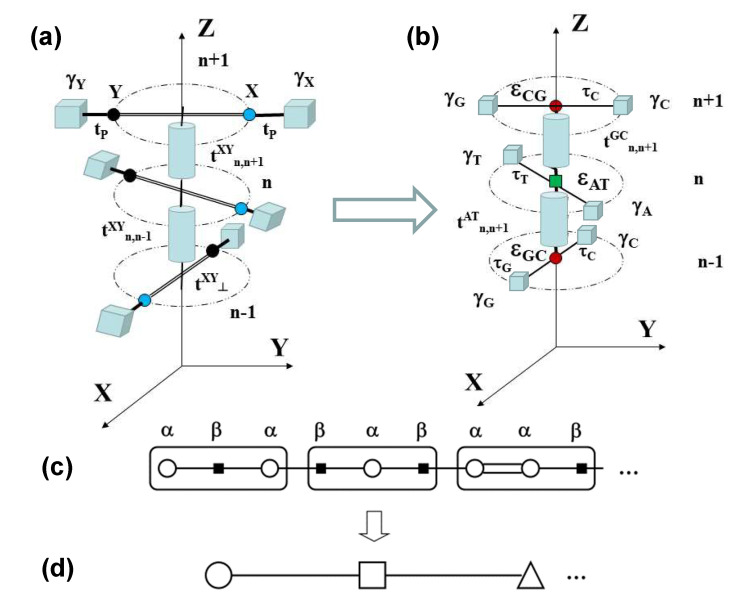
Sketch illustrating the three-step renormalization process mapping the 3D dsDNA electronic model shown in (**a**) into the 3D fishbone model shown in (**b**), then into the effective 1D diatomic and polyatomic lattice models displayed in (**c**) and (**d**), respectively, the latter corresponding to a chain made of codon triplet building blocks.

**Figure 4 materials-13-05119-f004:**
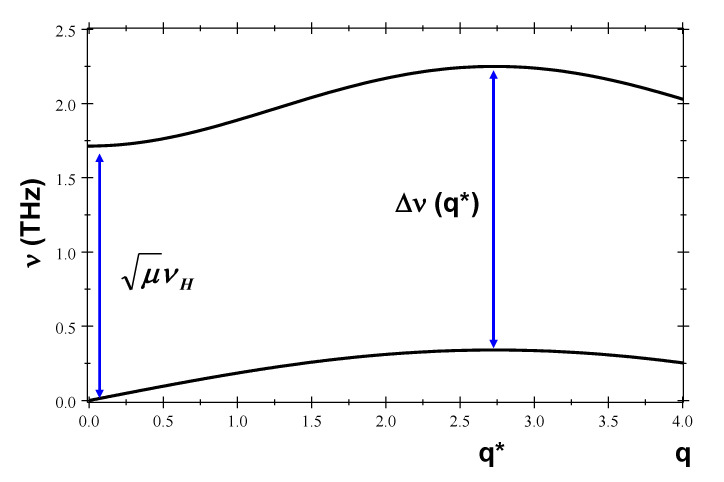
Acoustic and optical phonon branches for polyG-polyC and polyA-polyT dsDNA homopolymers (their $\nu_{-}\left(q\right)$ and $\nu_{+}\left(q\right)$ curves respectively coincide up to the third decimal place), where *q* is measured in nm${}^{-1}$. We used the characteristic and coupled frequency values listed in Table 2.

**Figure 5 materials-13-05119-f005:**
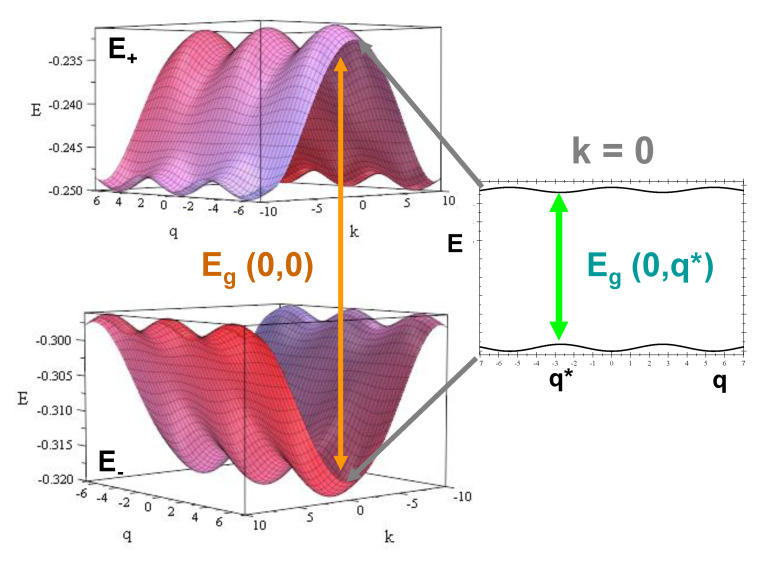
Electronic band structure for a polyG-polyC homopolymer. We used the electronic model parameter values listed in Table 1 and the choice γ=0 eV. The wavevectors *k* and *q* are measured in nm${}^{-1}$, and $q^{*}=\pi /\xi ≃2.732$ nm${}^{-1}$. The band structure of a polyA-polyT homopolymer is very similar (see Table 3) with the center of mass of the bands shifted upwards in energy by about 0.02 eV.

**Table 1 materials-13-05119-t001:** Geometrical, dynamical, lattice model, and electronic model parameters adopted in the dsDNA homopolymer model studied in this work. The same effective Morse potential is used to describe H-bonding in both GC and AT bps. The spring constant $k_{B}$ is difficult to estimate, and different possible values, ranging from 0.04 to 0.5 eV Å${}^{-2}$, have been reported in the literature [50,51,52,53,54,55].

Geometrical	Dynamical	Lattice	Electronic (eV)	Electronic (eV)
$\theta_{0}=\pi /5.2$ rad	M=653.5 amu	b=0.5 Å${}^{-1}$ [25,56]	$t_{⊥}^{GC}=0.01$ [38,57,58]	$\varepsilon_{G}=7.8-8.2$ [39,59]
$h_{0}=0.34$ nm	$\lambda_{GC}=1.130$	α=5 Å${}^{-1}$ [24,56,60]	$t_{⊥}^{AT}=0.02$ [38]	$\varepsilon_{A}=8.2$ [59]
$l_{0}=0.68$ nm	$\lambda_{AT}=1.028$	D=0.15 eV [30,56,60]	$t_{0}^{GG}=0.08$ [38,59,61]	$\varepsilon_{C}=8.9$ [59]
$R_{0}=1.00$ nm	$\mu_{GC}=4.015$	$k_{S}=0.7$ eV Å${}^{-2}$ [30]	$t_{0}^{AA}=0.09$ [38,58,59]	$\varepsilon_{T}=9.0-9.1$ [39,59]
ξ=1.15 nm	$\mu_{AT}=4.001$	$k_{B}=0.04$ eV Å${}^{-2}$ [56]	$t_{P}=1.5$ [62]	0≤γ≤12 [14,62]

**Table 2 materials-13-05119-t002:** Characteristic frequencies and scale times in the lattice dynamics of dsDNA polyG-polyC homopolymers. The $\nu_{\varphi \rho }$, $\nu_{\varphi S}$, and $\nu_{HS}$ values for polyA-polyT homopolymers are slightly smaller by just about 4 GHz.

Oscillation	$\omega_{k}$	($10^{12}$ rad s${}^{-1}$)	$\nu_{k}$ (THz)	$\tau_{k}$ (ps)
Twist	$\omega_{\varphi }$	1.07	0.17	5.87
Stacking	$\omega_{S}$	3.26	0.52	1.93
H-Bonding	$\omega_{H}$	5.34	0.85	1.18
Twist-Stacking	$\omega_{\varphi S}$	6.52	1.04	0.96
Twist-Radial	$\omega_{\varphi \rho }$	12.53	2.00	0.50
Stretch-Stacking	$\omega_{HS}$	14.13	2.25	0.45

**Table 3 materials-13-05119-t003:** PolyG-polyC and polyA-polyT band structure properties (measured in meV). We adopted the values γ=0, $a^{GC}=16.6$ eV, $a^{AT}=17.4$ eV, $b^{GC}=30.9$, and $b^{AT}=33.6$.

Homopolymer	$W_{+}$	$W_{-}$	$E_{g}\left(0,0\right)$	$E_{g}\left(0,q^{*}\right)$
PolyG-PolyC	29.7	24.4	80.3	73.0
PolyA-PolyT	22.0	16.6	93.5	87.4

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
