# Peer review of "Base-Pairs’ Correlated Oscillation Effects on the Charge Transfer in Double-Helix B-DNA Molecules"

_materials, 2020, doi:10.3390/ma13225119_

Round 1

Reviewer 1 Report

Wow! The manuscript is excellent. It is extremely thorough and scientifically sound. My research interested are in Chemical Physics/Physical Chemistry, and I enjoyed it greatly. With that said, I do believe there is room for improvement, and revisions should be considered before acceptance.

The main issue I have with the manuscript is that many will likely find it difficult to read and follow. The results will be of interest to a broad audience. But I fear that many will become disinterested when reading and stop. I do not suggest that the technical content be changed at all. But perhaps add material throughout the derivation to remind the reader of the motivation, and where you are going. Or, perhaps an overview paragraph could be useful in the Introduction to help the reader through the derivation part. Such a section might actually allow a reader to skip the derivation and still appreciate the work.

Again, I reiterate that I very much enjoyed the manuscript. My criticism is from an interested reader who would like to see the impact or your work increased if possible. 

Author Response

Thank you very much for your interest in my work and for your comments aimed at improving the readability of the manuscript. Following your suggestion I have added a full paragraph in the introductory section (lines 62-81) in order to help the reader through the derivation part.  

Reviewer 2 Report

Base-pairs Correlated Oscillation Effects on the Charge Transfer in Double-Helix B-DNA Molecules

Review Report

The manuscript is discussing the oscillation effects on the charge transfer in DNA molecule. I do believe that is an excellent subject to be discussed and is going to give citations to the journal as it is something that scientists could use to move further their studies. Despite the fact that this work seems promising to be applied in biological molecules, I would like to see how the author would approach the correlated oscillation effects when the DNA molecule is bonded with another ligand. How these oscillations will affect the charge when other small molecules are in present. This is something that will give a new perspective to the manuscript and will help other scientists to apply the author’s technique. Regarding the references, are well chosen having recent bibliographic data and quality references. The manuscript generally speaking it is well written and the theory is explained adequately. The use of English language is at high standard and the figures are in a good quality.

Best regards

Author Response

Thank you very much for your interest in my work and for your comments aimed at widening the audience of the manuscript. Following your suggestion I have added a full paragraph in the concluding section (lines 206-229) describing the effects related to the presence of small molecules bonded to the DNA one on the correlated oscillations. Some words on a similar role stemming from epigenetic processes (i.e., methylation) have been included as well. Three new references (82-84) have been included in the reference list.